# Pleiotropic Long-Term Effects of Atorvastatin on Posttraumatic Joint Contracture in a Rat Model

**DOI:** 10.3390/pharmaceutics14030523

**Published:** 2022-02-26

**Authors:** Erik Wegner, Ekaterina Slotina, Tim Mickan, Sebastian Truffel, Charlotte Arand, Daniel Wagner, Ulrike Ritz, Pol M. Rommens, Erol Gercek, Philipp Drees, Andreas Baranowski

**Affiliations:** Biomatics Group, Department of Orthopaedics and Traumatology, University Medical Centre of the Johannes Gutenberg University, 55131 Mainz, Germany; erik.wegner@unimedizin-mainz.de (E.W.); slotina.ekaterina@gmail.com (E.S.); tim.mickan@googlemail.com (T.M.); sebastian-truffel@t-online.de (S.T.); charlotte.arand@unimedizin-mainz.de (C.A.); daniel.wagner@unimedizin-mainz.de (D.W.); ulrike.ritz@unimedizin-mainz.de (U.R.); prommens@uni-mainz.de (P.M.R.); erol.gercek@unimedizin-mainz.de (E.G.); philipp.drees@unimedizin-mainz.de (P.D.)

**Keywords:** joint contracture, myofibroblast, atorvastatin, posttraumatic joint stiffness (PTJS), antifibrotic drugs, rat model, fibrosis, inflammation

## Abstract

The antifibrotic effect of atorvastatin has already been demonstrated in several organ systems. In the present study, a rat model was used to investigate the effect of atorvastatin on posttraumatic joint contracture. Forty-eight Sprague Dawley rats were equally randomized into an atorvastatin group and a control group. After initial joint trauma, knee joints were immobilized for intervals of 2 weeks (n = 16) or 4 weeks (n = 16) or immobilized for 4 weeks with subsequent remobilization for another 4 weeks (n = 16). Starting from the day of surgery, animals received either atorvastatin or placebo daily. After euthanasia at week 2, 4 or 8, joint contracture was determined, histological examinations were performed, and gene expression was assessed. The results suggest that the joint contracture was primarily arthrogenic. Atorvastatin failed to significantly affect contracture formation and showed a reduction in myofibroblast numbers to 98 ± 58 (control: 319 ± 113, *p* < 0.01) and a reduction in joint capsule collagen to 60 ± 8% (control: 73 ± 9%, *p* < 0.05) at week 2. Gene expression of α-smooth muscle actin (*α-SMA*), collagen type I, transforming growth factor β1 (*TGF-β1*) and interleukin-6 (*IL-6*) was not significantly affected by atorvastatin. Atorvastatin decreases myofibroblast number and collagen deposition but does not result in an improvement in joint mobility.

## 1. Introduction

Fibroproliferative disorder (FPD) presents in various forms. Common to all is an imbalance between the synthesis and degradation of connective tissue, the former being favored as a result of dysregulated and continuous inflammation [1]. In terms of posttraumatic joint stiffness (PJTS), chronic inflammation is initiated by acute or repetitive trauma. Hence, PJTS can affect most joints exposed to a traumatic stimulus. Though the progression of this condition is not life-limiting as it is for other systemic FPDs, structural and biomechanical alterations to the joint cause restricted joint motion and chronic pain. Depending on the joint, these symptoms can yield excruciating consequences, affecting the patient’s quality of life [1,2].

Despite substantial improvements regarding the pharmacotherapy of many systemic FPDs, PJTS’s own pharmacotherapy remains in its infancy. Today, treatment basically rests on three pillars: prolonged physical therapy, mobilization under general anesthesia and surgical removal of excess joint tissue. All of these measures potentially improve the range of motion (ROM) of the affected joint; however, the latter two may fuel the vicious circle by exacerbating the progression of tissue damage [3,4]. With none of the treatments working in a preventative manner, prophylactic pharmacotherapy needs to address the initial formation of excessive connective tissue. As for other FPDs, antifibrotic drug therapy may be a promising and novel constituent in the limited weaponry against PJTS [5].

On a cellular level, tissue-remodeling myofibroblasts are the key players in FPDs [1]. Within the connective tissue, myofibroblasts’ large focal adhesions enable cell-to-cell and direct contact with the extracellular matrix (ECM). Through these adhesions, mechanical tissue load can be sensed or applied by myofibroblasts [5,6]. The cell’s potent contractile activity is realized by the TGF-ß1-dependent formation of robust stress fibers containing α-SMA [6,7]. Myofibroblasts also account for permanent tissue retraction by abundant ECM deposition and complex collagen crosslinking [8,9]. Intriguingly, the mechanical load that these cells apply to the connective tissue leads to an additional release of ECM-bound TGF-ß1, perpetuating a profibrotic feedback loop [10]. Usually, myofibroblasts are present in negligible numbers in healthy tissue. However, PJTS is associated with markedly elevated cell counts [11,12,13]. Moreover, the aforementioned feedback loop results in self-maintaining and locally persistent myofibroblasts that circumvent apoptosis in affected tissue [10,14]. The myofibroblast population in FDS is as heterogenic as its progenitor cells. The progenitor depends on the organ affected, the underlying disorder, timing and the distinct signaling pathway. Tissue-specific fibroblasts and pericytes, however, are the mainstays of myofibroblast origin [15,16]. However, resident mesenchymal stem cells are increasingly recognized as another major contributor [15]

Myofibroblast differentiation is triggered primarily in response to TGF-ß1 with the assistance of other inflammatory cytokines, such as interleukin-1ß (IL-1ß) and IL-6 [11,17]. In other words, myofibroblasts are the key players, but TGF-ß is the master regulator of FPDs [1,18]. 

Numerous in vitro and some animal studies have not only proven that modification of profibrotic mediators and their downstream signaling cascade potentially interferes with the proliferation and differentiation of myofibroblasts but also regulates the cell’s ECM synthesis and ECM maturation [9,14,17].

Atorvastatin has a well-known effect on cholesterol levels as a reversible and competitive inhibitor of 3-hydroxy-3-methyl-glutaryl-coenzyme A reductase (HMG-CoA reductase). Additionally, an antifibrotic effect seems to be among its versatile pleiotropic properties [19,20]. Reduced synthesis of profibrotic mediators is accomplished partly by inhibiting small G-protein prenylation, thus inactivating the Ras signaling pathway [21,22]. As in the following two examples, the aforementioned effects were repeatedly demonstrated in FPDs of different organs and various experimental setups: dose-dependent administration of atorvastatin in cultured hepatic stellate cells noticeably attenuated myofibroblast cell differentiation, reduced the expression levels of profibrotic cytokines and decreased the production of collagens and α-SMA [23]. Similarly, atorvastatin prevented myofibroblast differentiation in fibroblastic cells specific to the human lungs when exposed to TGF-ß [24]. 

To clarify the relationship between oral atorvastatin treatment and PTJS development and severity, we undertook a blinded and randomized placebo-controlled trial in rats. Our objective was to determine if atorvastatin treatment is associated with less contracture formation. The primary efficacy endpoint was the reduction in joint contracture. Secondary endpoints were differences in cell numbers, changes in the morphology of the posterior joint capsule and alterations in gene expression of *TGF-β1*, *IL-6*, *α-SMA* and collagen type I. Our hypothesis was that atorvastatin reduces posttraumatic joint contracture via downregulation of profibrotic pathways and subsequent reduction in myofibroblast numbers. This study helps to elucidate some of atorvastatin’s antifibrotic mechanisms by evaluating its microscopic and macroscopic effects in the context of PJTS. 

## 2. Materials and Methods

### 2.1. Animals and Study Design

Experiments were carried out on 10-week-old male Sprague Dawley rats (n = 48) (Janvier Labs, Saint-Berthevin Cedex, France) with a mean weight of 393 ± 4 g. Animals were housed individually at room temperature in a 12:12 h light–dark cycle. Rats were randomly allocated into an atorvastatin group (n = 24) and a placebo group (n = 24) by computer-assisted group randomization (v 1.2, http://randomisation.eu, accessed on 12 January 2022). Knee joints were immobilized either for 2 weeks (subset 1, n = 16) or 4 weeks (subset 2 (n = 16) and subset 3 (n = 16)) after joint trauma. Animals in subset 1 were euthanized after 2 weeks, and animals in subset 2 were euthanized after 4 weeks. Animals in subset 3 underwent reoperation at week 4 with Kirschner wire (K-wire) removal. Subsequent remobilization with free cage activity was allowed until euthanasia at week 8 (Figure 1).

Sample size calculations were based on our previous study [25]. Regardless of the group, all animals underwent the same surgical procedure. Rats received either atorvastatin (15 mg/kg/day orally, Pfizer Pharma GmbH, Berlin, Germany) or a placebo (Winthrop Arzneimittel GmbH, Frankfurt am Main, Germany) once a day for 2, 4 or 8 weeks from operation day until euthanasia. The placebo group served as a control. CO_2_ asphyxiation served as the method of euthanasia. Atorvastatin dosage was adjusted to the current literature recommendations [23,26]. White chocolate spread was used as a vehicle for administration of ground atorvastatin or placebo. Additionally, standard rodent chow and water were provided ad libitum. The food vehicle ensured entire and immediate drug consumption. To rule out experimental bias, drug administration and surgical intervention were performed in a blinded manner. This study was approved by the local ethics committee of Rhineland-Palatinate (ID 23 177–07/G 13–1-043). Approval was granted on 7 August 2013 and renewed on 19 April 2016.

### 2.2. PTJS Model and Surgical Procedure

Joint trauma was induced according to our standard protocol, as previously described in detail [25,27]. In summary, the left knee of the anesthetized rats was hyperextended to −45° in order to disrupt the posterior joint capsule. After prepping, animals were operated on the left limb under sterile conditions. A 2 mm thick and 4 mm deep hole was drilled into the lateral femoral condyle (cartilage was meticulously spared from trauma). Further temporary K-wire transfixation of the adjacent load-bearing long bones in flexion of the knee joint was applied for 2 or 4 weeks and contributed to the formation of PTJS. 

General anesthesia was also used for K-wire removal in animal groups submitted to remobilization (subset 3) [25]. K-wire position and accidental fractures were ruled out by lateral radiography of the limb (MX-20 cabinet X-ray system, Faxitron, DOM 2009, Tucson, AZ, USA). 

Knees and elbows have been shown to be the most vulnerable joints to PJTS. A flexion contracture model was chosen to mimic PJTS in the knee, the most debilitating and clinically relevant form [17,28]. In particular, in a flexion contracture, the posterior capsule has proven to be at the heart of this condition with its densely packed connective tissue, whereas its anterior counterpart remains largely unaffected [29,30].

### 2.3. Knee Joint Angle Measurements

The deviation between the physiological extension angle of the animal’s knee joint prior to surgery and after immobilization or remobilization was defined as overall joint contracture. Joint contracture was divided into myogenic and capsular components. The myogenic component of joint contracture was defined as the difference between the joint angle before and after periarticular myotomy. The arthrogenic component was calculated by first adding a physiological myogenic extension inhibition of 21.7° to the baseline joint angle [25]. This value corresponds to the physiological baseline joint angle without muscles. The difference between the joint angle after periarticular myotomy and the baseline joint angle without muscles indicates the true arthrogenic contracture. Joint contracture was assessed under general anesthesia after K-wire removal. A custom-built joint fixation unit and an extension torque gauged to 35 N mm allowed for precise and standardized measurements, as previously reported in detail [25]. ROM was radiographically documented (MX-20 cabinet X-ray system, Faxitron, DOM 2009). Images acquired were processed and analyzed with ImageJ (v1.50). 

### 2.4. Tissue Preparation

For further histological evaluation and morphological assessment, knee joints were entirely removed. Fixation was achieved by immediate transfer of the tissue specimens into neutral buffered 4.5% formalin solution (Carl-Roth, Karlsruhe, Germany) for 48 h. Subsequently, the tissue was decalcified in 17.7% ethylenediaminetetraacetic acid (EDTA) (Applichem, Darmstadt, Germany) and tris(hydroxymethyl)aminomethane (TRIS) (Applichem, Darmstadt, Germany) buffered solution. Incubation for 6 weeks at room temperature produced optimal results. Then, tissue specimens were embedded in paraffin. Joint tissue was cut into 5 µm sagittal sections with a microtome. Only knee joint slices from the paramedian to mid-region were submitted for further evaluation. Hence, the following anatomical structures had to be included in a single slice: (I) femoral condyle, (II) tibial plateau, (III) triangular cross-section of the posterior part of the meniscus, (IV) adjacent posterior joint capsule and (V) synovial folds. Slices with tissue-related artifacts and torn structures were excluded. Following standard protocol, sections were stained with hematoxylin and eosin (H&E) for morphometrical evaluation. For distinct visualization of connective tissue, Elastica van Gieson staining was performed in accordance with the manufacturer’s instructions (Merck Millipore, Burlington, VT, USA). For additional immunohistological assessment, slices were submitted to the following protocol: Prior to staining tissue, specimens were deparaffinized and rehydrated. Antigen sites were revealed by incubation of the tissue with proteinase K (Dako, Hamburg, Germany) for 10 min. After repeatedly washing the slices with phosphate-buffered saline (PBS) solution, 3% H_2_O_2_ (Merck, Darmstadt, Germany) was added for another 10 min to block endogenous peroxidase activity. Before adding 10% horse serum for 30 min, tissue specimens were washed with the aforementioned buffered solution again. Sequentially, tissue was incubated with the α-SMA secondary antibody overnight at 4 °C (Arigo Biolaboratories, Hsinchu City, Taiwan). Alternatively, controls were treated with 1% bovine serum albumin (BSA) in PBS buffer for the same duration. Followed by multiple washing steps before and after applying biotinylated link (Dako), tissue specimens were incubated with horseradish peroxidase-conjugated streptavidin (HRP, Dako). Specimens were washed. Subsequently, samples were exposed to 3,3′-diaminobenzidine (DAB) solution (Dako) until sufficient color staining was achieved. The reaction was immediately stopped by adding aq. dest. Finally, hematoxylin was used for counterstaining. 

### 2.5. Tissue Morphometrics and Histology

For subsequent evaluation, pictures of obtained sections were taken and processed with ImageJ (v1.50). Assessments were performed by two blinded and independent investigators. Generally, the assessments reflect the average of four tissue sections per subset (Figure 1). Since one histological specimen from placebo subset 1 could not be evaluated (due to incorrect treatment of the tissue) and one animal from placebo subgroup 3 had to be euthanized early, the total sum of evaluable sections was n = 10 in the placebo group and n = 12 in the atorvastatin group. As previously described, morphometric measurements consist of a superior and an inferior subdivision of the posterior synovial length and the capsular thickness behind the posterior border of the meniscus [27].

For histological evaluation of the joint capsule, an area of 97,035.9 μm^2^ (361.4 μm × 268.5 μm) adjacent to the posterior border of the meniscus was defined for cell counting. This area is referred to as the staining area. Threshold limits were adjusted to a cell diameter (artifact avoidance) for ImageJ-assisted counting in binary color converted images. For Elastica van Gieson staining, binary-colored images were adjusted to a color threshold. Color ratios were determined and compared. In regard to myofibroblast discrimination, α-SMA can also be expressed in pericytes or smooth muscle cells situated in vascular walls [31]. Therefore, α-SMA-stained cells in close vicinity to luminal structures were excluded, as they may resemble the latter two cell types. Staining areas were set in relation to the total area of the tissue section. 

### 2.6. Tissue Preparation and Quantitative PCR

Knee joint capsules (n = 12 in the control; n = 11 in the atorvastatin group) were immediately stabilized in RNAlater (Thermo Fisher Scientific, Waltham, MA, USA) and stored at –20 °C until further processing. Tissue homogenization was performed in two consecutive steps. After specimens were manually ground in a mortar using liquid nitrogen, the obtained granulates were suspended in TRIzol (Thermo Fisher Scientific) before submitting them to Precellys homogenizer (Bertin Technologies, Montigny-le-Bretonneux, France). Homogenization was performed according to the manufacturer’s recommendations. Subsequent ribonucleic acid (RNA) extraction was performed after a standard phenol–chloroform protocol using the supernatants of the centrifuged homogenates (Sigma-Aldrich, St. Louis, MO, USA). Residual RNA pellets were washed twice with 75% ethanol (EtOH) before diluting RNA pellets in nuclease-free water (Sigma-Aldrich). Photometric quantitation of the isolates was measured at 260 nm with NanoDrop (Thermo Fisher Scientific). For conversion of RNA (0.8 µg/sample) into complementary deoxyribonucleic acid (cDNA), the isolated RNA was treated with DNAse before reverse transcription was performed using and M-MuLV reverse transcriptase, Random Primer Mix (New England Biolabs, Ipswich, MA, USA) and nucleotides (Bioron GmbH, Ludwigshafen, Germany). Specific primer sequences (Table 1) for real-time polymerase chain reaction (qPCR) were designed using the manufacturer’s web-based designing tool (Eurofins Scientific, Luxembourg City, Luxembourg) in accordance with the sequences provided by National Center of Biotechnology Information’s database on nucleotides (https://www.ncbi.nlm.nih.gov/nucleotide/, accessed on 12 January 2022). Glyceraldehyde 3-phosphate dehydrogenase (GAPDH) served as the endogenous control and was used to normalize gene expression. qPCRs were run on a qTOWER3 (Jena Analytik, Jena, Germany) using the Biozyme Blue S’Green qPCR Master Mix (Biozyme Scientific GmbH, Hessisch Oldendorf, Germany) according to the manufacturer’s recommendations and the following cycling conditions: 2 min at 95 °C, 2 min at 95 °C, and (5 s at 95 °C, 30 s at 60 °C) × 40. For calculation of qPCR results, Δ-cycle threshold (ΔCt) values were used. Gene expression levels of *TGF-β1*, *IL-6*, *α-SMA* and collagen type I under atorvastatin treatment were compared to the gene expression of the control in the tissue of the posterior joint capsule.

### 2.7. Statistical Analysis

Statistical analysis was conducted with the SPSS 28.0.1.0 (142) software (SPSS Inc., Chicago, IL, USA) and GraphPad Prism 9.1.2 (226) software (GraphPad Software, San Diego, CA, USA). Quantitative results are presented as medians and quartiles or as means ± standard deviation. Joint angles and capsular parameters were analyzed with the Shapiro–Wilk test for normality assumption, followed by Levene’s test for the assumption of homogeneity of variances and one-way ANOVA, and gene expression was compared with multiple unpaired t-tests (comparison of ΔCt values). Measurements were carried out in triplicates. A *p*-value < 0.05 was considered statistically significant. In box-and-whisker plots, the whiskers extend to the highest and lowest values. A line across the box indicates the median.

## 3. Results

### 3.1. Perioperative Weight Development and Complications

At the beginning of the study, the average rat weight was 396 ± 19 g in the control (n = 24) and 390 ± 16 g in the atorvastatin group (n = 24, Figure 2). Weight increased by 10–49 g every week. A slight decrease in weight from 528 ± 27 g to 525 ± 20 g was only observed in the atorvastatin group at week 4 after K-wire removal. This evened out towards the end of the study, and the weights of the rats no longer differed between groups (control: 607 ± 32 g (n = 7); atorvastatin: 600 ± 27 g (n = 8)). No rat required early euthanasia due to postoperative weight loss. However, one rat in the control group (subset 3) suffered an intraoperative fracture after week 4 and was euthanized.

### 3.2. Development of Posttraumatic Joint Contracture

#### 3.2.1. Overall Joint Contracture

We compared joint angles of the knee joints at different time points with the respective baseline value of each rat to obtain the extent of posttraumatic joint contracture (Figure 3). In both groups, the majority of the contracture formed after 2 weeks and was 60.9° ± 11.4° in the control and 68.5° ± 15.7° in the atorvastatin group (*p* < 0.01 vs. respective baseline value; n.s. between treatment groups). The development of contracture slowed down between weeks 2 and 4 and increased only to 72° ± 9.7° (control) and 76.3° ± 12.5° (atorvastatin), respectively. After K-wire removal at week 4, a significant reduction in contracture was observed in both groups, leading to an improvement of 33.6° ± 11.3° in the control and 27.2° ± 13.9° under atorvastatin treatment (*p* < 0.01). However, we found no difference in the severity of contracture between the two treatment groups at any time point.

#### 3.2.2. Myogenic Component of Joint Contracture

By measuring the increase in joint angle after periarticular myotomy, we determined the muscular contribution to joint contracture. It was found that myogenic joint contracture was most pronounced after 2 weeks (control: 29.7° ± 5.9°; atorvastatin: 37.5° ± 17.4°), then initially decreased slightly until the 4th week of immobilization (control: 23.5° ± 14.0°; atorvastatin: 16.8° ± 10.6°), and then further decreased to 20.4° ± 8.1° in the control and 12.7° ± 6.3° in the atorvastatin group after remobilization (Figure 4). Although no significant differences were detectable between the treatment groups at any time point (*p* = 0.15 at week 8), a significant decrease in myogenic contracture was observable for atorvastatin between week 2 and weeks 4 (*p* < 0.01) and 8 (*p* < 0.01).

#### 3.2.3. Arthrogenic Component of Joint Contracture

To determine the capsular component of joint contracture, we measured the joint angle in extension after myotomy and calculated the residual contracture compared with the baseline value (Figure 5). In both groups, the greatest increase in contracture to 52.9° ± 14.0° and 52.7° ± 15.2° (control vs. atorvastatin, respectively) was measured after 2 weeks of immobilization. Further immobilization until the 4th week resulted in a smaller increase in contracture: by 12.6° to 65.5° ± 18.3° in the control (*p* = 0.627) and by 28.1° to 80.8° ± 20.8° in the atorvastatin group (*p* < 0.05). Remobilization from the 4th to the 8th week reduced the contracture by 28.9° to 36.7°± 16.2° in the control (*p* < 0.05) and by 43.9° to 37.0° ± 10.6° in the atorvastatin group (*p* < 0.001)), respectively. The differences between the treatment groups were not statistically significant at any time point. The tendency towards a greater increase in contracture in the atorvastatin group after 4 weeks (*p* = 0.36 vs. control) was compensated by a more pronounced improvement in mobility under remobilization, and by week 8, no difference in capsular contracture was detectable.

### 3.3. Histological Changes in the Joint Capsule

#### 3.3.1. Joint Capsule Length and Diameter

The diameter of the posterior joint capsule increased during the study period in both treatment groups (Table 2). In the atorvastatin group, there was an increase from 0.7 to 1.3 mm (*p* < 0.01) between weeks 2 and 4, with no further gain in thickness after remobilization until week 8. In the control, the initial increase in diameter was slower, but there was a clear tendency for further growth after remobilization up to a diameter of 1.6 mm (*p* = 0.10), without significant differences between treatment groups (*p* = 0.24, 0.12 and 0.87 vs. atorvastatin at weeks 2, 4 and 8).

We observed a positive correlation between the capsule length and the duration of the study (Table 2). The length of the superior capsule increased on average by only 1 mm between weeks 2 and 8, while the length of the inferior capsule increased by between 1.6 mm (atorvastatin) and 3.0 mm (control) during the same period. Under atorvastatin treatment, inferior capsule length tended to be less than that of the control at each time point (*p* > 0.05), whereas superior capsule length was nearly identical between groups.

#### 3.3.2. Numerical Proportion of Effector Cells and Collagen Deposition

Immunohistochemical staining revealed α-SMA-positive myofibroblasts, which could be distinguished from α-SMA-positive endothelial cells of the vessel walls and from α-SMA-negative cells (Figure 6). 

After exclusion of the vessel-associated cells, α-SMA-positive cells were expressed as a ratio to the total cell number in the same histological section (Figure 7). Examination of the histological sections of the posterior joint capsule showed that atorvastatin treatment reduced myofibroblast numbers at week 2 (n = 98 ± 58) in comparison to the control (n = 319 ± 113, *p* < 0.01). Differences in myofibroblast numbers between groups leveled off by weeks 4 and 8 and were no longer significantly different. We found myofibroblast numbers of 239 ± 169 at week 4 and 163 ± 95 after remobilization at week 8 in the atorvastatin group. In the control, we found myofibroblast numbers of 156 ± 95 and 79 ± 28 at weeks 4 and 8, respectively. This fact is also reflected in the cell number ratios when the number of myofibroblasts at the different time points is related to the total number of cells (Figure 7).

The amount of collagen type I was determined to assess the degree of fibrosis of the posterior capsule. Atorvastatin lowered collagen deposition in the posttraumatic joint capsule to 60 ± 8% versus 73 ± 9% in the control (*p* < 0.05) at week 2, but not at weeks 4 and 8 (67 ± 10% and 72 ± 6% vs. 63 ± 9% and 76 ± 12% in the control, respectively; Figure 8).

### 3.4. Alterations in Gene Expression in the Joint Capsule

During the first two weeks of immobilization after trauma, no differences in gene expression of the proinflammatory markers *TGF-β1* and *IL-6* were detected (Figure 6). The reduction in myofibroblast numbers (Figure 7) and collagen content in the posterior capsule under atorvastatin was not reflected in a corresponding reduction in gene expression of *α-SMA* and collagen type I at week 2 (Figure 9a,b). Whereas a trend toward the upregulation of gene expression of *TGF-β1*, *α-SMA*, collagen type I and *IL-6* under atorvastatin was evident after 4 weeks of immobilization, a trend reversal was observed after remobilization at week 8. At that point, a trend toward the downregulation of expression by atorvastatin, particularly that of *TGF-β1* and *α-SMA*, was noticeable (Figure 9a,c).

## 4. Discussion

In the present randomized blind study, we investigated the effect of atorvastatin on the development of posttraumatic joint contracture of the knee joint in 48 Sprague Dawley rats over a total period of 8 weeks. The main part of the contracture developed during the first two weeks after injury while the joint was immobilized. As expected, the increase in contracture slowed by week 4. This rapid development of contracture was previously demonstrated by Wilson et al. in their early study on joint contractures in rats [32]. The arthrogenic component of the contracture was largely responsible for the limitation of joint mobility throughout the study (Figure 5), whereas the myogenic component was within (control) or below (atorvastatin) the physiological range of uninjured limbs (Figure 4). These findings are in accordance with the observations of Dunham et al., who found a contribution of only 10% of muscles and tendons to elbow contracture in a rat model after 6 weeks of immobilization [33]. In our previous study utilizing the same rat model of posttraumatic knee contracture, the physiological myogenic limitation of extension of the knee joint averaged 21.7° in rats [25]. In their study of immobilization-induced knee contracture in rats without previous trauma, Trudel et al. demonstrated a myogenic contracture of 10°/30% at week 4 [34]. This direct comparison between atraumatic and posttraumatic contracture suggests that musculature appears to play a lesser role in posttraumatic joint contracture. 

After remobilization of the joint by K-wire removal, a significant improvement in range of motion was observed in both groups from week 4 to week 8 (Figure 2). Since the myogenic component no longer accounted for a significant proportion of the total contracture after 4 weeks of immobilization, the gain in joint mobility due to remobilization is almost exclusively attributable to a reduction in the arthrogenic component of the contracture. In fact, we even observed that the myogenic component dropped below physiological levels after remobilization (Figure 4). Therefore, the remaining contracture can be ascribed solely to the arthrogenic component (Figure 5). Consistent with our results, Dunham et al. reported that the myogenic component of the persistent contracture decreased to 0% after 6 weeks of remobilization [33].

Despite remobilization, contractures of 33.6 ± 11.3° in the control and 27.2° ± 13.9° under atorvastatin treatment remained in the current study (Figure 3). Four-week joint immobilization without prior trauma showed an improvement in overall joint contracture to 13° in a similar rat model after subsequent four-week remobilization [35,36]. In a direct comparison between our study and the aforementioned atraumatic rat model, the posttraumatic contracture improved to a greater degree (by 38.4° (control) and 49.1° (atorvastatin)) than the atraumatic contracture (by 31°) with four weeks of remobilization [35,36]. However, due to the more pronounced contracture after 4 weeks of immobilization, a stronger residual contracture remained in the posttraumatic contracture even after this improvement.

There are no comparative studies on the effect of atorvastatin on joint contractures. In the present study, remobilization showed a significantly stronger effect on the recovery of joint mobility than the administration of atorvastatin (Figure 3). While no influence on the arthrogenic contracture component could be demonstrated under atorvastatin treatment (Figure 5), there was a clear tendency to reduce the myogenic contracture component compared with the control (Figure 4). However, numerous studies have demonstrated the antifibrotic effect of atorvastatin in various organ systems, such as the liver [37,38,39,40]. In the development of posttraumatic joint contracture, fibrosis of the joint capsule develops with an increased amount of type I collagen and a rise in the number of myofibroblasts, which reaches its peak 2 weeks after trauma [12,41]. Atorvastatin reduced the myofibroblast count by 75% after 2 weeks compared with the control (Figure 7). Accordingly, we demonstrated that collagen deposition in the joint capsule was significantly reduced by atorvastatin after 2 weeks. At the later time points of the study, there was no pronounced difference in myofibroblast numbers between the groups, but the number of myofibroblasts correlated positively with overall joint contracture and capsular joint contracture during the immobilization period (Figure 3, Figure 5 and Figure 7). The fact that a severe arthrogenic contracture had developed in both groups after 2 weeks, although the number of myofibroblasts could be significantly reduced under atorvastatin, indicates posttraumatic scarring without a relevant influence of myofibroblasts. It must be assumed that due to the serious initial joint injury with capsular rupture, a cascade with excessive healing and scarring had started, which could not be alleviated even by the reduction in the myofibroblast number under orally administered atorvastatin treatment. Nonetheless, in their recently published study in a similar rat model, Mao et al. were able to improve knee joint ROM by 41° after 6 weeks of immobilization by daily intraperitoneal injection of the TGF-β receptor I kinase inhibitor LY2157299 compared with vehicle-treated rats [42]. In addition to the possibly better efficacy of a drug, its form of administration may well play a role. In the recent literature on drug therapy for joint contracture, intra-articular administration of a drug seems to be tested much more frequently and may be more effective than oral administration [43,44,45,46]. We chose to test orally administered atorvastatin because daily intra-articular injections are inferior to oral administration of a drug in clinical use. This is due to their invasiveness, potential complications, such as joint infections or bleeding, and reduced patient compliance with injections. In fact, the authors are aware of only one study of posttraumatic joint contracture in a rat model in which oral administration of a drug was tested: Efird et al. demonstrated a 12° improvement in ROM two weeks after trauma and immobilization after daily oral administration of the leukotriene receptor antagonist montelukast [47]. In another study of posttraumatic joint contracture, albeit in a rabbit model, Li et al. found a 93.6° improvement in joint contracture at 30 days with twice-daily oral administration of celecoxib [48]. We observed a positive correlation of capsule length and diameter with the duration of the study, without demonstrating a clear effect of atorvastatin on these capsule parameters. We had hypothesized that joint mobility would be positively correlated with capsular length and negatively correlated with capsular diameter, but this did not prove to be the case (Table 2, Figure 3). Mao et al., on the other hand, found a correlation between decreasing capsular thickness and increased joint mobility. However, in the same study, the joint capsule did not show a significant increase in length with improved ROM [42].

The development of posttraumatic joint disorder is significantly mediated by the proinflammatory cytokines TGF-β1 and IL-6, among other discussed factors [49,50]. The resulting inflammation leads to myofibroblast proliferation and fibrosis of the joint capsule [51,52]. Although the differences between the control and the atorvastatin group were not found to be statistically significant, gene expression of the inflammatory cascade as well as of collagen type I and *α-SMA* correlated with the severity of arthrogenic contracture under atorvastatin treatment (Figure 9). A tendency to upregulate gene expression of the mentioned factors under atorvastatin treatment at week 4 was reflected in a tendency to a more pronounced arthrogenic contracture, whereas downregulation of those genes resulted in a slight improvement of arthrogenic contracture at week 8 (Figure 5 and Figure 9). Mao et al. were able to significantly decrease the gene expression of *TGF-β1*, *α-SMA* and *COL-1A1* by intra-articular administration of LY2157299 [42]. Morrey et al. hypothesized that in the complex process of fibrosis development, downregulation of profibrotic genes and upregulation of alternative pathways could be thwarted by intersecting feedback loops. In their study, they found no biomechanical effect on joint contracture after intra-articular injection of fosaprepitant (a selective NK-1 antagonist) despite the upregulation of antifibrotic genes [53].

The present study has some limitations. In two recently published studies on posttraumatic contracture in rats, the influence of cartilage and ligaments was described both during immobilization and especially during remobilization [33,54]. Dunham et al. stated that ligaments and cartilage are mostly responsible for the motion loss after remobilization and that the contribution of the capsule to joint contracture accounts for only 26% after 6 weeks of remobilization. This represents a confounding factor that should be considered in future studies. In our study, no dissection was performed beyond the capsular boundary, which could have been used to distinguish between the different arthrogenic parameters of the contracture. It is possible that the initial trauma in the current study was too severe and led to a particularly pronounced healing response with exuberant fibrosis that could not be mitigated even by an effective agent. Future work will focus on testing additional oral drugs for the prophylaxis of posttraumatic joint disorder. The drugs to be tested should be able to inhibit multiple pleiotropic pathways that function to activate profibrotic genes. Their influence on ligamentous and chondral factors will also be investigated in more detail in an optimized rat model.

## 5. Conclusions

In the present study of the pleiotropic effect of atorvastatin on posttraumatic joint contracture, we observed a histologically significant reduction in myofibroblast numbers and collagen deposition by atorvastatin in the posterior joint capsule in the early phase (2 weeks). Gene expression of *IL-6*, *TGF-β1*, *α-SMA* and collagen type I tended to be upregulated after 4 weeks under atorvastatin treatment and slightly decreased after another 4 weeks of remobilization compared with the control without statistical significance. However, these observations did not translate into a corresponding improvement in joint mobility. Lastly, the myogenic component was observed to have little importance as a contributor to the degree of PTJS. Despite the good effects at the cellular level, treatment of PTJS with orally administered atorvastatin does not seem to be an option because of the negligible effects on ROM. In the search for alternative therapies of PTJS, future studies should also try to elucidate the redundant pathways of posttraumatic fibrosis.

## Figures and Tables

**Figure 1 pharmaceutics-14-00523-f001:**
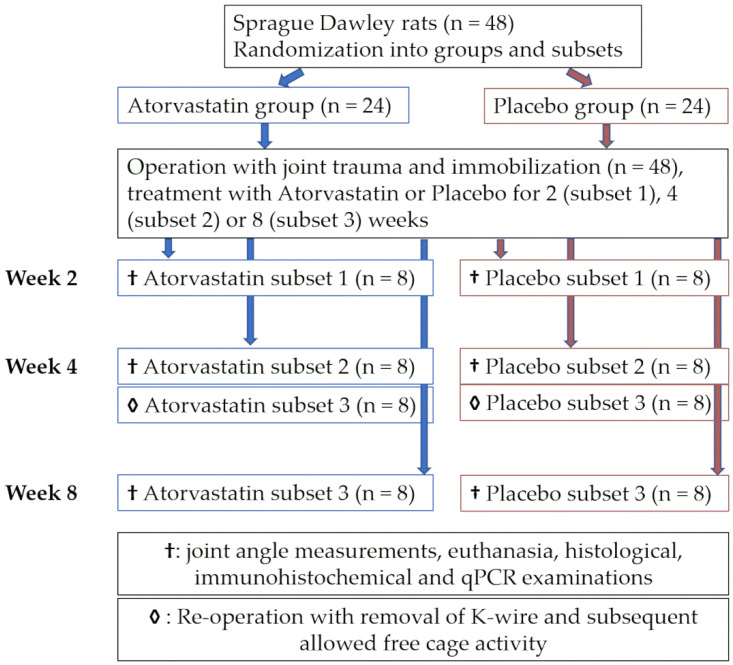
Animal allocation to atorvastatin and placebo groups and subsets.

**Figure 2 pharmaceutics-14-00523-f002:**
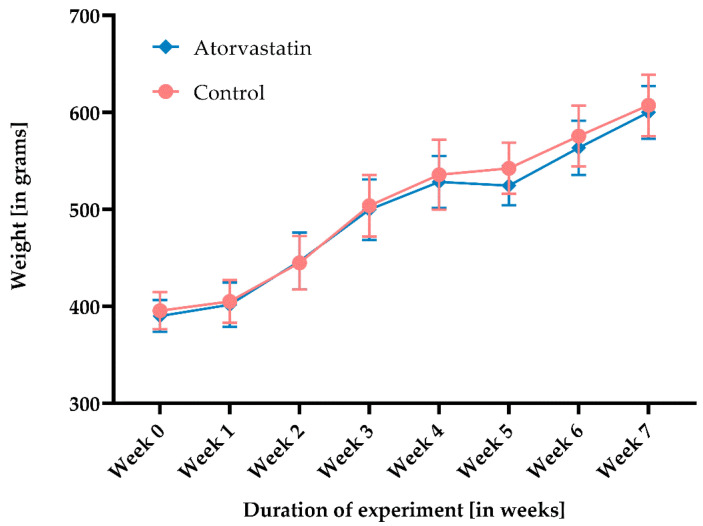
Weight development of rats in the course of the study without differences between atorvastatin and the control (total number: n = 48 by the end of week 2, n = 32 by week 4, and then n = 15).

**Figure 3 pharmaceutics-14-00523-f003:**
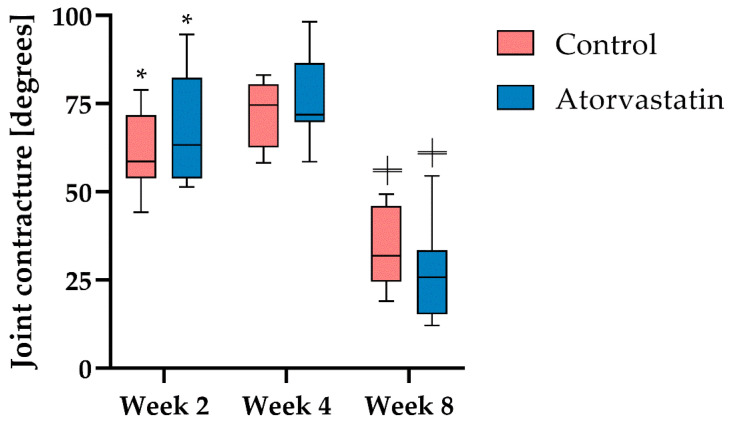
Overall joint contracture after immobilization at week 2 (n = 8 per group) and week 4 (n = 8 per group) and after remobilization at week 8 (n = 8 (atorvastatin); n = 7 (control)). Data are presented as boxplots with median, minimum and maximum. Significant differences to the respective baseline value are marked with * (*p* < 0.05), and significant differences to the contracture at week 4 are indicated with ╪ (*p* < 0.01).

**Figure 4 pharmaceutics-14-00523-f004:**
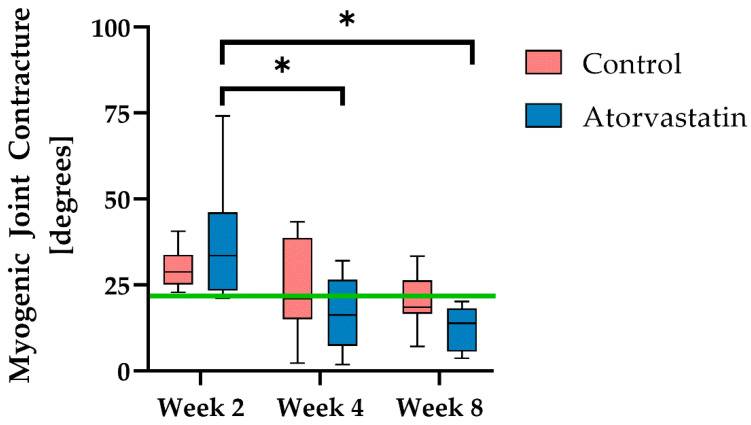
Myogenic joint contracture at weeks 2, 4 (immobilization) and 8 (remobilization); n = 8 per group and time point, and n = 7 only in control after 8 weeks. Data are presented as boxplots with median, minimum and maximum. The green line represents the physiological muscular inhibition of joint extension in rats. Significant differences (*p* < 0.01) are marked with *.

**Figure 5 pharmaceutics-14-00523-f005:**
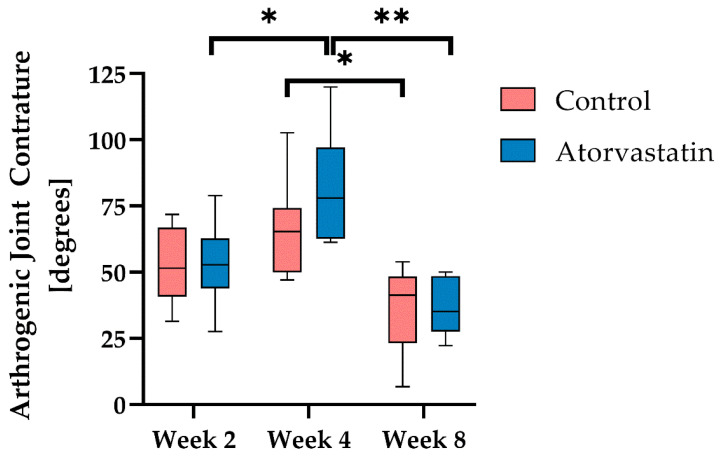
Arthrogenic joint contracture at weeks 2, 4 (immobilization) and 8 (remobilization); n = 8 per group and time point, and n = 7 only in control after 8 weeks. Data are presented as boxplots with median, minimum and maximum (*: *p* < 0.05, **: *p* < 0.001).

**Figure 6 pharmaceutics-14-00523-f006:**
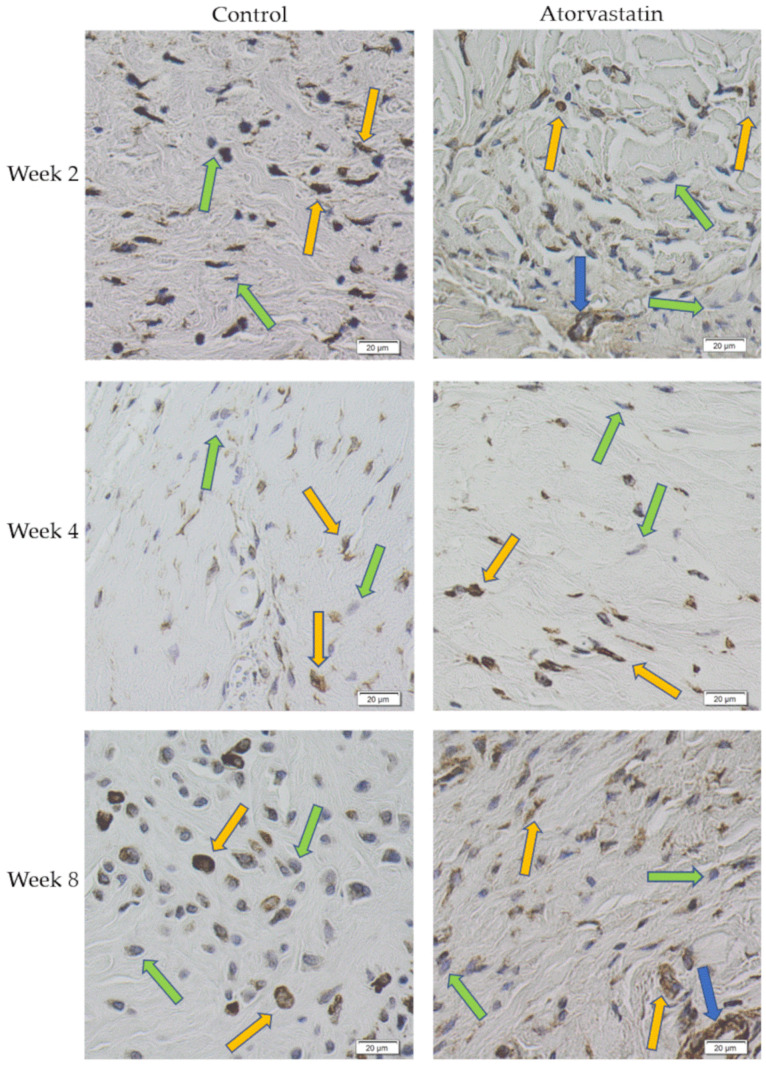
*α*-SMA-positive staining of myofibroblasts (yellow arrows) and vascular endothelial cells (blue arrows); α-SMA-negative staining of fibroblasts (green arrows).

**Figure 7 pharmaceutics-14-00523-f007:**
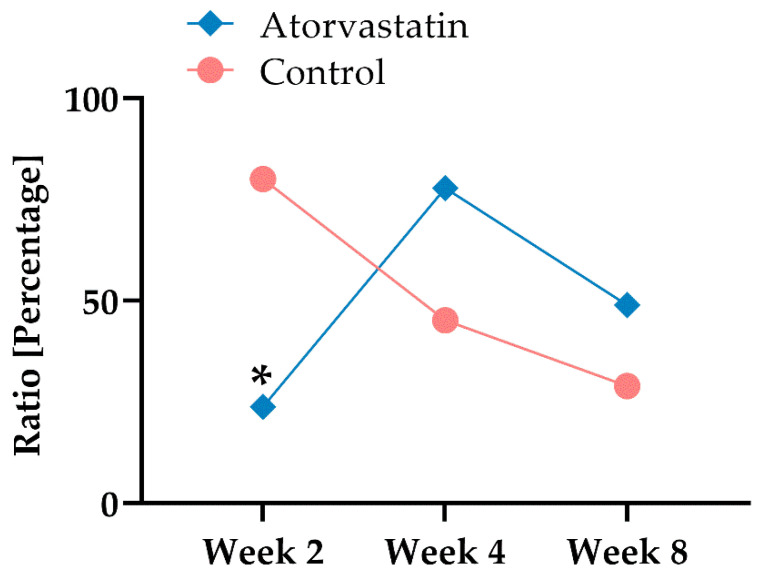
Percentage of myofibroblasts in relation to the total cell count of the posterior capsule (*: *p* < 0.01 to respective control).

**Figure 8 pharmaceutics-14-00523-f008:**
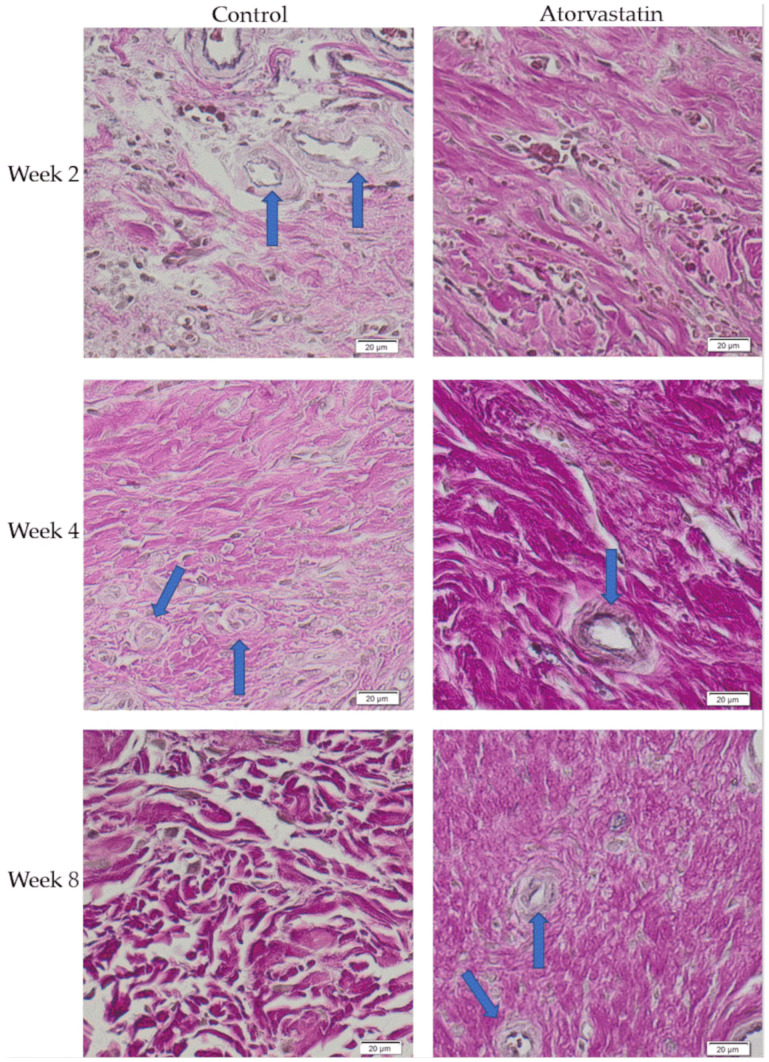
Elastica van Gieson staining of the posterior joint capsule. Collagen is shown in pink, and blood vessels are shown with a blue arrow.

**Figure 9 pharmaceutics-14-00523-f009:**
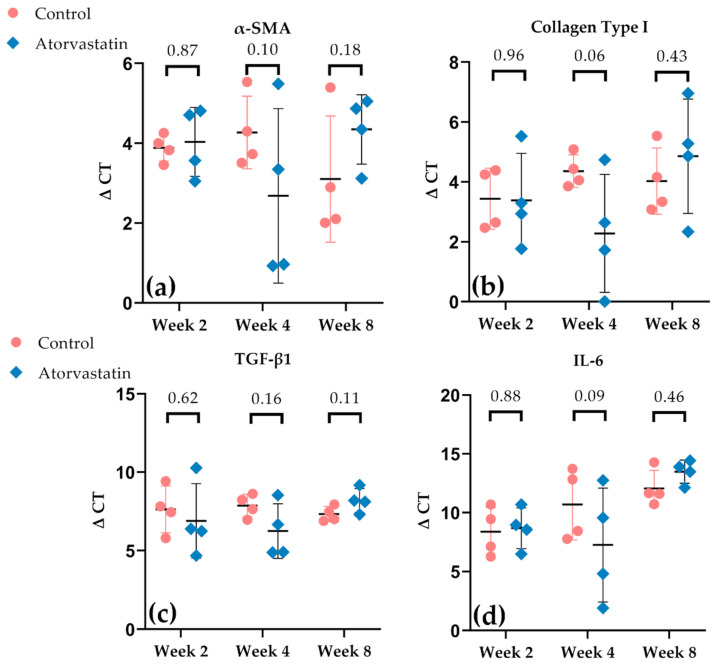
Gene expression of *α-SMA* (**a**), collagen type I (**b**), *TGF-β1* (**c**) and *IL-6* (**d**) is depicted as ΔCt values in scatter dot blot design with mean and SD. Differences between atorvastatin and control are expressed as *p*-values above the scatter dots.

**Table 1 pharmaceutics-14-00523-t001:** Genes and primer sequences used for qPCR.

Gene	Sequence	Primer
*GAPDH*	Forward	AACGACCCCTTCATTGACCT
	Reverse	CCCCATTTGATGTTAGCGGG
*TGF-β1*	Forward	CCCTACATTTGGAGCCTGGA
	Reverse	CGCACGATCATGTTGGACAA
*IL-6*	Forward	CCACCCACAACAGACCAGTA
	Reverse	ACTCCAGAAGACCAGAGCAG
*α-SMA*	Forward	CATCATGCGTCTGGACTTGG
	Reverse	CCAGGGAAGAAGAGGAAGCA
Collagen type I	Forward	CCCCAAATGCTGCCTTTTCT
	Reverse	CTGGGTAGGGAAGTAGGCTG

**Table 2 pharmaceutics-14-00523-t002:** Length and diameter of posterior joint capsule (presented as mean and standard deviation in mm).

Time Since Operation	Capsular Length, Superior (in mm)		Capsular Length, Inferior (in mm)		Capsular Diameter (in mm)	
	Atorvastatin	Control	Atorvastatin	Control	Atorvastatin	Control
Week 2	2.0 ± 0.9	1.7 ± 1.6	0.9 ± 0.9	1.1 ± 0.4	0.7 ± 0.1	0.8 ± 0.6
Week 4	2.2 ± 0.2	2.2 ± 0.7	1.6 ± 0.8	1.9 ± 1.3	1.3 ± 0.3	1.0 ± 0.2
Week 8	3.0 ± 1.0	2.7 ± 0.6	2.5 ± 1.9	4.1 ± 2.1	1.3 ± 0.2	1.6 ± 0.1

## Data Availability

Data are available on request.

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
