# Peer review of "Pleiotropic Long-Term Effects of Atorvastatin on Posttraumatic Joint Contracture in a Rat Model"

_pharmaceutics, 2022, doi:10.3390/pharmaceutics14030523_

Round 1
Reviewer 1 Report
This is an interesting and well written paper that fits the Journal scope. Methods are adequate and well explained. Results are clearly presented and well discussed. Referencing to previous literature is adequate as well. I have no hesitation to recommend straightforward publication.
Only, please enlarge the scale bar in Figure 5a – it is really difficult to see.
Reviewer 2 Report
Manuscript “Pleiotropic long-term effects of atorvastatin on posttraumatic joint contracture in a rat model”.
The topic is interesting. However, there are several issues that the authors should address to improve their study.
My comments are as follows:
- Line 63-64: “Myofibroblasts derive from fibroblasts and a variety of other cells. All of them serve as myofibroblast precursors.” This part should be better explained and references should be added.
- At the end of the introduction, the authors should report the aim/s of the study.
- Section 2.1: Study design should be better explained. A scheme illustrating how the authors divided the rats in the different groups/treatments would help the reader. It is unclear if the authors started to treat the rats with atorvastatin/placebo during the immobilization period. Joint trauma should be better defined. Supplier of atorvastatin should be specified. It is unclear what did the authors use as placebo.
- Lines 174-175: if the tissues morphometric analysis were performed by two blindere investigators, it is unclear why all the assesments reflected the average of four tissue section in the Atorvastatin and only three in the control group.
- Line 179: why did the authors select an area of 97,035.9 μm2? “μm2” should be corrected.
- Lines 203-205: it is unclear if the authors treated the isolated RNA with DNAse in order to eliminate possible DNA contamination before performing cDNA trasnscription.
- Line 210: “rRNA” is unclear. Did the authors mean mRNA?
- Figure 1: how many rats were included? The authors are referring to the 4 week immobilization group but it is unclear if these rats received either atorvastatin or placebo once a day for 2, 4 or 8 weeks from operation.
- Figure 2-3-4: Since the allocation of rats in experimental design is unclear, there is some confusion here as well. Specifically, are the rats shown at 2 weeks different from those shown at 4 weeks or are they the same? I supposed that they are the same. However, it seems that they are different by reading the methods (n=16 two weeks and n=32 four weeks). How many? Was treatment (placebo/atorvastatin) started immediately upon immobilization?
- Why did the authors focused on posterior joint capsule? The rationale should be explained.
- Figure 5: Could the authors report representative IHC images of atorvastatin and control group at 2, 4 and 8 weeks?
- Lines 358-361: Could the authors report representative images of collagen type I IHC of atorvastatin and control group at 2, 4 and 8 weeks?
- The conclusions should be improved. It seems that atorvastatin is not useful for the treatment of PJTS.
Minor comments:
- Lines 58-61: the sentences are repeated twice.
- Line 438: a full-stop is missing.
- Contracted forms (“isn’t”) should be not used.
- Abbreviations should be defined at first mention.
- Symbols for genes should be italicized.
Reviewer 3 Report
This study analyzes some of Atorvastatin’s antifibrotic mechanisms by closing ranks between its microscopic and macroscopic effects in the context of PJTS. In the present study of the pleiotropic effect of atorvastatin on posttraumatic joint contracture, the authors observed histologically a significant reduction in myofibroblast number and collagen deposition by atorvastatin in the posterior joint capsule in the early phase . Gene expression of IL-6, TGF-β1, α-SMA, and collagen type I tended to be upregulated after 4 weeks under atorvastatin and slightly decreased after another 4 weeks of remobilization compared with control without statistical significance
The methodology is exhaustive and focused on demonstrating the proposed hypothesis. All methods are adequately described and reproducible
The results are described based on the experiments carried out
The discussion is clear, commenting on the published bibliography
Author Response
We thank the reviewer for the positive evaluation of our manuscript.
Round 2
Reviewer 2 Report
The manuscript improved.